# Would *Cutibacterium acnes* Be the Villain for the Chronicity of Low Back Pain in Degenerative Disc Disease? Preliminary Results of an Analytical Cohort

**DOI:** 10.3390/jpm13040598

**Published:** 2023-03-29

**Authors:** Vinícius Magno da Rocha, Carla Ormundo Gonçalves Ximenes Lima, Gustavo Baptista Candido, Keila Mara Cassiano, Kai-Uwe Lewandrowski, Eliane de Oliveira Ferreira, Rossano Kepler Alvim Fiorelli

**Affiliations:** 1Department of General and Specialized Surgery, School of Medicine, Federal University of the State of Rio de Janeiro (Unirio), Rio de Janeiro 21941-901, Brazil; 2Department of Medical Microbiology, Institute of Microbiology Paulo de Góes, Federal University of Rio de Janeiro (UFRJ), Rio de Janeiro 21941-901, Brazil; 3Niterói D’Or Hospital, Niterói 24230-251, Brazil; 4Department of Statistics, Institute of Mathematics, Fluminense Federal University (UFF), Niterói 24220-008, Brazil; 5Center for Advanced Spine Care of Southern Arizona, Tucson, AZ 85712, USA; 6Surgical Institute of Tucson, Tucson, AZ 85711, USA

**Keywords:** intervertebral disc, low back pain, discogenic pain, chronic pain, *Cutibacterium acnes*, neuropathic pain

## Abstract

In the last decade, several studies have demonstrated *Cutibacterium acnes* colonization in intervertebral discs (IVDs) in patients with lumbar disc degeneration (LDD) and low back pain (LBP), but the meaning of these findings remains unclear. Being aware of this knowledge gap, we are currently conducting a prospective analytical cohort study with LBP and LDD patients undergoing lumbar microdiscectomy and posterior fusion. The IVDs samples collected during the surgeries are subjected to a stringent analytical protocol using microbiological, phenotypic, genotypic, and multiomic techniques. Additionally, pain-related scores and quality-of-life indexes are monitored during patient follow-up. Our preliminary results for 265 samples (53 discs from 23 patients) revealed a *C. acnes* prevalence of 34.8%, among which the phylotypes IB and II were the most commonly isolated. The incidence of neuropathic pain was significantly higher in the colonized patients, especially between the third and sixth postoperative months, which strongly suggests that the pathogen plays an important role in the chronicity of LBP. The future results of our protocol will help us to understand how *C. acnes* contributes to transforming inflammatory/nociceptive pain into neuropathic pain and, hopefully, will help us to find a biomarker capable of predicting the risk of chronic LBP in this scenario.

## 1. Introduction

Low back pain (LBP) is a highly prevalent complaint among the population. It has several consequences for the patient and for society, such as physical and psychological limitations, drug abuse, absenteeism from work, and the potential to overload social security and health systems [1,2,3]. It is a multifactorial etiological condition whose incidence is about five times higher in patients with lumbar disc degeneration (LDD) [4,5,6,7,8,9].

In the last decade, advances in molecular identification and microbiological culture techniques have redirected the researchers’ attention to the isolation of low-virulence pathogens in different musculoskeletal conditions, such as aseptic failures of surgical implants and intervertebral discs (IVDs) degeneration [10,11,12,13,14,15,16,17,18]. Among these pathogens, the one that is the most isolated in patients with LDD is *Cutibacterium acnes*, which has a prevalence rate of around 40% [19,20,21]. Given the public health importance of LBP and its high association with LDD, there has been extensive research to understand the role of *C. acnes* in this scenario.

Studies evaluating Modic changes (MCs) are the most common in relating LDD to *C. acnes* colonization [10,22,23,24,25]. MCs represent fissures in the endplates of the vertebrae and are common magnetic resonance imaging (MRI) findings in patients with LDD [26,27]. Substantial recent evidence has correlated MCs with IVD colonization by *C. acnes*, but these findings are not unanimous among studies [19,28,29]. Rajasekaran et al. applied proteomic techniques to demonstrate how host defense proteins and other proteins are indicative of bacterial viability and proliferation in IVDs colonized by *C. acnes*. These authors integrated the presence of the pathogen with mechanical factors, such as the occurrence of fissures in the endplates, within what they called the unified hypothesis of disc degeneration [30]. Capoor et al. demonstrated the positive effect of *C. acnes* in IVD cell cultures on the expression of IL-1β and other host defense molecules related to the neo-innervation of the annulus fibrosus and the degradation of the extracellular matrix (ECM) of IVDs [31]. Other authors have indicated that such responses were conditioned by the immune profile of the host and the virulence mechanisms of the *C. acnes* strain; some biofilm-producing phylotypes use this feature to remain inert to defense cells within IVDs, while others produce ECM degradation enzymes in IVDs and activate inflammatory cascades through the Toll-like receptor system [31,32,33,34,35,36].

Although there is growing evidence pointing to the active role of *C. acnes* in IVD de-generation pathways and the activation of inflammatory cascades, so far, the available literature has not fully explained the mechanisms related to the generation and/or chronification of LBP in patients with LDD. There is a dearth of studies that aim to monitor the evolution of patients’ pain complaints, especially after IVD collection and not just at the time of enrollment for microbiological investigation. There are no standardized or reproducible protocols, so the data produced by analytical cohorts can barely be used in trustworthy reviews and meta-analyses. There are also few alternatives to compensate for the lack of “normal” control IVDs that, despite bacterial colonization, present the physiological changes of aging, which will vary according to genetics and an individual’s choices throughout their life.

Aware of these limitations, our research group designed a prospective cohort study for the follow-up of patients with LBP and LDD. In addition to the control of clinical variables before and after sampling (surgical procedure in which the IVD specimens are obtained), the design of this cohort also houses an analytical study that brings together microbiological, molecular, and multiomic techniques in an unprecedented way [37]. This protocol was registered at ClinicalTrials.gov, has been validated, and is already in progress.

Given the topic’s relevance, this manuscript aims to present the preliminary micro-biological results of this research and to correlate them with the clinical profile of the patients being followed up.

## 2. Materials and Methods

This cohort study integrates clinical and analytical data and is in progress at the Spine Surgery Center of São Matheus Hospital, in partnership with the Gaffrée and Guinle University Hospital and the Laboratory of Anaerobic Biology, at the Institute of Microbiology Paulo de Góes (IMPG) of the Federal University of Rio de Janeiro (UFRJ).

Patients with LBP for at least 3 months and findings confirming LDD on MRI were recruited and selected for surgical treatment through microdiscectomy and lumbar fusion. The procedures were performed in a single hospital by the same team of surgeons. All patients included in the study were informed about the research and freely agreed to participate before signing the Informed Consent Form for the study. The selection criteria for the study are listed in Table 1.

The patients were monitored by a team of specialists in pain and/or spinal surgery for at least 3 months before surgery and for the first 6 months after surgery. During patient recruitment and follow-up, we collected demographic, clinical, and imaging data (Table 2). Only the group of researchers involved with the laboratory analyses had access to the microbiological results before the end of the 6th postoperative month.

Intravenous antibiotic prophylaxis was performed exclusively during the surgery for all the patients (cefazolin 1 g, repeated every 4 h for the duration of the surgery), as is the hospital protocol of cutaneous asepsis using 2% chlorhexidine gluconate degerming solution and 0.5% alcoholic solution of chlorhexidine gluconate.

During surgery, five tissue fragments were collected for each IVD addressed. A set of sterile surgical tweezers was used for collection, with one set being used for each tissue fragment. Immediately after collection, the clamp with the sample was handed over to a member of the laboratory who was also present in the operating room during the procedure. Each sample was inoculated into a flask containing thioglycolate culture medium (Merck, Rio de Janeiro, Brazil) containing 20 glass beads in each tube. After sampling, the vials containing the clinical specimens were kept at room temperature and were transported to the laboratory of Biology of Anaerobes at the Federal University of Rio de Janeiro (UFRJ) (Instituto de Microbiologia Paulo de Góes- IMPG, Medical Microbiology Department, UFRJ) within a maximum period of 2 h.

The tubes containing IVD fragments undergo vortexing (Even EVX2800-BI^®^, Curitiba, Brazil) for 15 s. All tubes were kept in a bacteriological incubator (from 35 °C to 37 °C), and after 24 h, the first subculture was performed. Using a 100 µL bacteriological loop, aliquots of thioglycolate were removed and seeded: one onto a blood agar base and another on anaerobic blood agar. The first plate was incubated under a capnophilic atmosphere, and a second plate was kept in strict anaerobiosis using an anaerobic jar or Glove Box (Coy Labs^®^, Grass Lake, MI, USA). The plates were read after 24 h, and, in case of growth, the colonies were identified in the MALDI-TOF MS (Bruker^®^, Billerica, MA, USA). In the absence of growth, they were reincubated, and a new reading was performed after 72 h. The same procedure was used for all of the thioglycolate vials (totaling 10 vials for each level approached). The thioglycolate tubes containing the clinical specimens were kept under incubation (35 °C to 37 °C) for at least 14 days. The subculture procedure was performed after 72 h and then every 3 days, with the last being performed on the 14th day of incubation in the same solid culture media and atmospheres used in the first 24 h.

The analyses of the samples were carried out sequentially according to the performance of the surgical procedures in the selected participants. Once the pathogen in the culture media has been identified, we proceed with its phenotypic and genotypic characterization by using MALDI-TOF MS (Bruker Biotyper^®^, Bremen, Germany) and r-16S DNA analysis using the Multiplex Touch-Down PCR technique (species-specific and Filotyping PCR), respectively.

We perform multiomic analyses for: (1) the identification of prokaryotic proteins that may be acting as molecular patterns associated with pathogens (PAMPs) or damages (DAMPs) in the host response (proteomics) and for (2) the definition of a metabolomic signature for patients with LBP who have LDD associated with *C. acnes* colonization (metabolomics). Variables distributions in the negative and positive to *C. acnes* groups were compared appropriately by the Fisher’s exact test or Mann–Whitney tests.

## 3. Results

At the time of writing this manuscript, the microbiological analyses of 265 lumbar IVD samples obtained from 23 patients (52.2% female and 47.8% male) with a mean age of 46.5 years had been completed (SD 10.2; median 44.0). No clinical disease was reported with a frequency greater than 50% (SAH, diabetes, endocrine, or vascular diseases). Obesity was identified in 43.5% of the patients (seven with grade I and three with grade II). Smoking was reported by 39.1% of the patients (consumption of at least one cigarette/day over the course of 1 year), and 60.9% declared themselves to have a sedentary lifestyle (light physical activity of less than 150 min per week). The patients’ baseline characteristics are listed in Table 3.

### 3.1. Microbiological Analyses

Culture results are available for the 265 IVD samples (a total of 53 discs collected in 23 patients). *C. acnes* was isolated in discs from 34.8% of the patients and more frequently among men (62.5%). The number of samples collected from each patient ranged from 5 to 20, totaling 265 samples, with an average of 11.5 samples collected per patient. In 160 samples from 15 patients, no positive samples were found. The 8 positive patients had 105 samples together, of which 79 were positive. Therefore, it is estimated that 24.8% of samples from positive patients do not show a positive result, showing the need to investigate several samples per patient (at least 1/0.752 = 1.3 samples from each patient). Therefore, it is recommended to collect at least 2 samples from each patient for this investigation. The number of positive samples from each positive patient ranged from 2 to 20, averaging 9.9 positive samples from each positive patient. Table 4 brings together the main statistics on the isolation of *C. acnes* from the IVD samples.

Phylotypes IB and II were found the most frequently in the colonized patients, being isolated in five and four patients, respectively; these two phylotypes jointly colonized 42.9% of the positive samples. The IA1 and IA2 phylotypes were isolated less frequently, being in mutual association in one patient and associated with the IB and/or II phylotypes in three other patients. In one of the patients colonized by *C. acnes*, phylotyping was not possible due to the death of the strains. Figure 1 graphically illustrates the distribution of the phylotypes in positive IVDs.

### 3.2. Preoperative Features

All of the patients had LBP (43.5% mechanical pain and 56.5% sciatica pain), and an indication for surgical intervention was the failure of conservative treatment. The mean duration of symptoms before surgery was 12.1 months (SD 7.4; median 12 months), which was higher among the patients with sedentary lifestyles, patients with SAH, and those with mechanical pain (*p*-values of Mann–Whitney test lower than 5%). None of the patients had a motor deficit, and neuropathic pain was only identified in one of them (score greater than 4 on the DN4 questionnaire) during the preoperative clinical evaluation. Regarding the baseline characteristics (age, BMI, smoking, sedentary lifestyle, and clinical comorbidities), there was no significant difference between the groups with positive and negative cultures (Mann–Whitney tests and Fisher’s exact tests comparing the distributions of the two groups with *p*-values > 0.05). There was also no significant difference between these two groups in the preoperative Oswestry Disability Index (ODI) and EuroQol (EQ-5D) Quality of Life Scores (Mann–Whitney tests with *p*-values > 0.05).

### 3.3. Operative Features

In all of the patients, the surgical technique used was the same (microdiscectomy and transforaminal posterior fusion), with only the number of IVDs addressed in each patient varying (mean of 2.3—SD 1.0; median 3.0). The mean duration of the procedures was 3.7 h (SD 1.6; median 4.0), with an average of 1.6 h/level (SD 1.7; median 0.4). Regarding the duration of surgery and the number of IVDs addressed per patient, there was no significant difference between the groups with positive and negative cultures in the disk (Mann–Whitney test with *p*-values > 0.05). There were no clinical or operative complications for the patients included in this part of the study.

### 3.4. Postoperative Features

The mean length of hospital stay was longer in the patients with positive IVD cultures (7.4 days—SD 3.0; median 6.5 days) than in the patients with negative cultures (5.8 days—SD 1.6; median 6.0), but this difference was not statistically significant (Mann–Whitney test with *p*-value of 0.190). Time of opioid usage was significantly higher in the colonized patients (Mann–Whitney test with *p*-value of 0.001), as was the time off work (Mann–Whitney test with *p*-value of 0.0038).

Despite an overall improvement in VNS scores after surgery, when comparing the pain levels in patients with and without positive cultures, this improvement was not significant before the third postoperative month (Mann–Whitney tests with *p*-values > 5%). However, the VNS distributions from the third to the sixth months show a significant difference in the observed improvement between the two groups, being smaller in the positive group (Mann–Whitney tests with *p*-values < 5%). *C. acnes* colonization also seems to have influenced the occurrence of neuropathic pain in the postoperative period, with a significantly higher incidence among the patients with positive cultures, as seen in Figure 2 (87.5% in the sixth postoperative month according to the DN4 questionnaire and Fisher’s exact tests with *p*-values < 0.05). The neuropathic pain statistics at different moments during clinical follow-up are displayed in Table 5.

According to the functional assessment using ODI, we noted that there was an improvement in the global scores between the preoperative period and the sixth month after surgery, but this evolution was not statistically significant between the groups with and without positive cultures (Fisher’s exact and Mann–Whitney tests with *p*-values > 0.05).

From the assessments of the quality of life using the EQ-5D, the surgical intervention had a significant and positive impact on all of the assessment domains except for the self-care domain (Wilcoxon tests with *p*-values < 0.05). However, comparing the groups, we noticed that the presence of bacteria did not significantly interfere with the benefit of treatment before the sixth postoperative month (Mann–Whitney test with *p*-values > 0.05). Table 6 compares the medians of the EQ-5D domains.

### 3.5. Modic Changes

Just over half of the patients had MCs on preoperative MRI in at least one of the surgically approached levels (52.2%). Among the colonized patients, a significant association was identified between *C. acnes* and the occurrence of Modic type 1, while in the group without Modic, the frequency of C. acnes was 0,0%; in the ones with Modic, 66.7% were positive for *C. acnes* (Fisher’s exact test with a *p*-value of 0.001).

## 4. Discussion

Our protocol revealed a *C. acnes* in disc prevalence (34.8%) and MC association (66.7%) close to the rates reported in the literature. The largest systematic review and meta-analysis ever published on the subject showed a prevalence of 28.7% in 1454 patients with disc degeneration [21]. This review gathered 14 studies whose methods varied both in sampling (number of samples per disc, column segment approached, and sampling technique), the culture media used, and the sample incubation time. This variance stems from the lack of a universal protocol for the isolation of *C. acnes* from IVD cultures, which hinders the performance of complementary analyses of the pathogen, such as strain phylotyping.

*C. acnes* has different and distinct evolutionary lineages, with the designated phylotypes IA1, IA2, IB, IC, II, and III being based on Multilocus Sequence Typing and Whole Genome Sequencing [38]. Each phylotype has a specific pathogenicity mechanism and produces virulence factors that, depending on the tissue they are colonizing, can trigger an inflammatory response or remain inert to the host’s immune system [34,39]. Rollason et al. highlighted the high proportion of phylotype II among positive IVD cultures from patients with LDD [40]; this phylotype produces the enzyme hyaluronate lyase (HYL), which degrades hyaluronic acid, an important component of ECM in IVDs, by contributing to its degeneration [41]. Moreover, hyaluronic acid degradation products have already been identified as activators of inflammatory pathways, with the production of molecules that integrate known discogenic pain pathways [42]. HYL has two variants, with the HYL-IB/II variant, which is found in phylotypes IB and II, being the most active [41,43]; phylotypes IB and II also secrete the cytotoxin CAMP-1 (Christie–Atkins–Munch–Petersen factor), which has a strong interaction with Toll-like receptor 2, amplifying the inflammatory response in the IVD microenvironment [33,44]. In an animal model study, Lan et al. demonstrated a positive effect of phylotypes IB and II on the expression of MMP-13 (Matrix Metallopeptidase 13), which is active in the degradation of collagen, an important component of the annulus fibrosus in IVDs [34]. Our preliminary results demonstrate a high frequency of phylotypes IB and II, which were isolated in association in 42.9% of the positive samples. Although the number of patients monitored at present limits our conclusions, the patients colonized by phylotypes IB and II had the worst VNS and ODI scores in the sixth postoperative month. Since the IVD is not entirely removed during surgery, the remaining tissue could continue to harbor *C. acnes* and participate in the LBP chronification process.

Unprecedentedly, our study continues to monitor patients with LDD after IVD collection, and among the differences observed in the first part of our results, the one that caught our attention the most was the change in the pain patterns presented by the patients over the postoperative period. Among the colonized patients who had mechanical and/or nociceptive pain in the preoperative period, 37.5% manifested characteristics of neuropathic pain up to the third month after surgery, with an increase in this proportion to 87.5% by the sixth postoperative month. These data should direct studies to the mechanisms through which *C. acnes* may be acting in the modulation of pain generation and chronicity mechanisms. *C. acnes* has already been linked to elevated levels of neurotrophins related to the penetration of nociceptive neurons into the deeper layers of the annulus fibrosus in cell cultures [32] as well as to the stimulation of the expression of IL-1β and other molecules associated with disc degeneration, inflammation, neo-innervation, and the activation of pain pathways [32,45,46,47,48]. Capoor et al. highlighted the secretion of hemolytic enzymes and pore-forming toxins by phylotypes IA, IB, and IC, which determines ionic influx and the chronic firing of action potentials in nociceptive neurons [49]. Freemont et al. demonstrated the presence of neurons that express the nocigenic Substance P in the internal annulus of the IVD and/or in the nucleus pulposus of patients with chronic LBP [50]. From these two authors’ works, it is inferred that *C. acnes* colonization of IVDs chronically exposes the newly formed nociceptive neuron endings to pain triggers, resulting in chronic LBP. Future studies on these and other possible mechanisms involved in the chronicity of LBP in this type of patient profile represent further challenges. Mapping the phylogenetic epidemiological profile of IVD colonization in LDD and monitoring the responses of the host’s immune system of the host for each manifested virulence factor are the next steps.

A variety of biological (e.g., genetics, immune response, infections, and age), psychological (e.g., depression, anxiety, and stress), and social (e.g., lack of monitoring, discrimination, and violence) factors are involved in conditions that generate chronic pain [51]. These factors determine changes in small molecules, known as metabolites, which can be identified and quantified using metabolomic techniques [52]. In a biological system, the metabolomic profile, that is, the type and number of metabolites in a sample, provides information on both the physiological and pathological processes of an individual [52,53]. In our study, based on the strains of isolated *C. acnes*, metabolomic analyses have been carried out to search for a metabolomic profile for chronic LBP related to the pathogen and LDD. Another multiomic analysis that has been carried out with isolated strains is proteomics analysis, whose main goal is to identify bacterial proteins in different IVD phylotypes that may be acting as inflammatory and nocigenic triggers.

The use of multiomic techniques, in addition to culture isolation techniques, is the most modern strategy for identifying biomarkers related to bacterial activity [51,54,55,56,57,58,59,60]. Summarizing the complex cellular mechanisms of restricted groups of molecules is an arduous task that involves numerous stages of analysis, data processing, and clinical validation of results. This task is even more difficult because of the global scarcity of research centers that gather such data. For the population of interest in this manuscript, the identification of a biomarker that could be non-invasively isolated from peripheral blood or urine, for example, would avoid the failure of surgical interventions, thus allowing for more effective treatment of LBP. This is indeed a strong justification for us to follow this protocol, encouraging us to share our results and to motivate the performance of other studies in this area, which is home to one of the most prevalent complaints in the areas of pain and spinal surgery. In conclusion, *C. acnes* is a pathogen that causes a low-grade symptom infection which is difficult to be diagnosed, especially in our country. This is the first study to demonstrate the isolation of *C. acnes* and phylotypes in the disc content of patients with LBP. With the increasing number of cases involving this pathogen, particularly in orthopedic surgeries related to nonspecific infection, low immune response, and chronic pain, there is an urgent need for new biomarkers to improve not only the presence of *C. acnes* but its relation to the infectious disease. As a result, the “omics” approach can aid in the development of new diagnostic approaches for detecting *C. acnes* and assisting clinicians in infection and chronic pain in their patients.

## Figures and Tables

**Figure 1 jpm-13-00598-f001:**
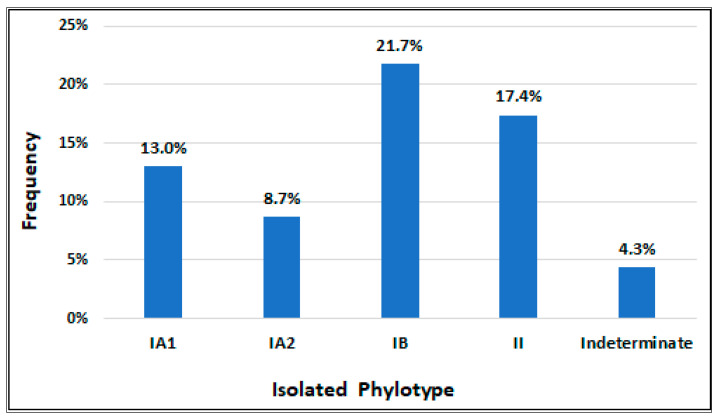
Frequency of *Cutibacterium acnes* phylotypes isolated.

**Figure 2 jpm-13-00598-f002:**
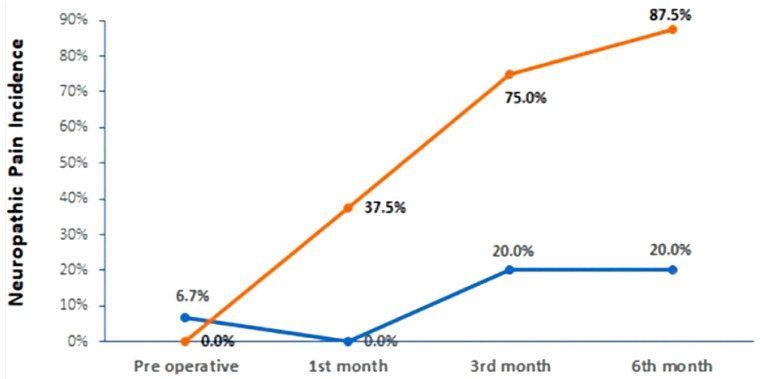
Evolution of neuropathic pain incidence on the groups of patients negative and positive for *Cutibacterium acnes***.** Frequency of Isolated Phylotypes.

**Table 1 jpm-13-00598-t001:** Selection Criteria of Patients.

Inclusion	Exclusion
Age between 18 and 65 years	History of previous lumbar spine surgery
LBP ^a^ lasting more than 3 months	Chemotherapy or pulse therapy with corticoids
MRI ^b^ performed less than 6 months before surgery	History of spinal infection
Failure of conservative treatment for at least 6 weeks	Previous intradiscal therapies

^a^—Low back pain; ^b^—Magnetic resonance imaging.

**Table 2 jpm-13-00598-t002:** Patients’ Collected Data.

Data	Moment of Collection			
	Enrollment.	1st Postop. Mo.	3rd Postop. Mo	6th Postop. Mo
Age, gender, BMI ^a^, No. of levels	X			
Sedentary lifestyle, smoking	X			
SAH ^b^, diabetes, other comorbities	X			
MRI ^c^ (Modic changes)	X			
VNS ^d^, DN4 ^e^, ODI ^f^	X	X	X	X
EuroQol questionnaire (EQ-5D)	X			X
Time off work		X	X	X

^a^—Body mass index; ^b^—Systemic arterial hypertension; ^c^—Magnetic resonance imaging; ^d^—Visual numeric scale; ^e^—Douleur neuropathique questionnaire; ^f^—Oswestry Disability Index.

**Table 3 jpm-13-00598-t003:** Baseline Patients’ Characteristics Means According to Microbiological Profile.

Variable	Category	Global (n = 23)	*C. acnes* (−) (n = 15)	*C. acnes* (+) (n = 8)	*p*-Value
Gender	Female	52.2%	60.0%	37.5%	0.400 ^a^
	Male	47.8%	40.0%	62.5%	
BMI	Normal weight	21.7%	20.0%	25.0%	0.728 ^b^
	Overweight	34.8%	40.0%	25.0%	
	Obesity I	30.4%	26.7%	37.5%	
	Obesity II	13.0%	13.3%	12.5%	
Comorbidities	Diabetes	47.8%	33.3%	75.0%	0.089 ^a^
	Obesity	43.5%	40.0%	50.0%	0.667 ^a^
	SAH ^c^	39.1%	40.0%	37.5%	1.000 ^a^
Bad Habits	Smoking	39.1%	26.7%	62.5%	0.179 ^a^
	Sedentary Lifestyle	60.9%	53.3%	75.0%	0.400 ^a^
Preop. Pain	Sciatica	56.5%	53.3%	62.5%	1.000 ^a^
	Mechanics	43.5%	46.7%	37.5%	
Preop DN4 ^d^	≤1	26.1%	26.7%	25.0%	0.899 ^a^
	2	47.8%	40.0%	62.5%	
	3	21.7%	26.7%	12.5%	
	≥4	4.3%	6.7%	0.0%	
Preop ODI ^e^	Minimal/Moderate	47.8%	53.3%	37.5%	0.294 ^b^
	Severe/Crippled	52.2%	46.7%	62.5%	0.667 ^a^

^a^—Fisher’s exact test; ^b^—Mann–Whitney test; ^c^—Systemic arterial hypertension; ^d^—Douleur neuropathique questionnaire; ^e^—Oswestry Disability Index.

**Table 4 jpm-13-00598-t004:** *Cutibacterium acnes* isolation statistics according to the number of patients included in the study.

Number of Patients	23
Number of evaluated discs	53
Number of discs samples	265
Number of *C. acnes* positive patients	8
Number of *C. acnes* positive discs	18
Number of *C. acnes* positive samples	79
Prevalence of *C. acnes* positive patients	34.8%

**Table 5 jpm-13-00598-t005:** Neuropathic Pain Evaluation (DN4 Score).

Variable	Category	Global (n = 23)	*C. acnes* (−) (n = 15)	*C. acnes* (+) (n = 8)	*p*-Value
Preoperative DN4	Absent	26.1%	26.7%	25.0%	0.776 ^a^
	Possible	21.7%	26.7%	12.5%	
	Probable	47.8%	40.0%	62.5%	
	Present	4.3%	6.7%	0.0%	
DN4 1st month	Absent	26.1%	26.7%	25.0%	0.265 ^a^
	Possible	47.8%	73.3%	0.0%	
	Probable	13.0%	0.0%	37.5%	
	Present	13.0%	0.0%	37.5%	
DN4 3rd month	Absent	43.5%	60.0%	12.5%	0.019 ^a^
	Possible	0,0%	0.0%	0.0%	
	Probable	17.4%	10.0%	12.5%	
	Present	39.1%	10.0%	75.0%	
DN4 6th month	Absent	60.0%	60.0%	0.0%	0.011 ^a^
	Possible	6.7%	6.7%	0.0%	
	Probable	13.3%	13.3%	12.5%	
	Present	20.0%	20.0%	87.5%	

^a^—Mann–Whitney test.

**Table 6 jpm-13-00598-t006:** EQ-5D Quality of Life Evaluation.

Domain	Assessment	Median Score in the Domain		
		*C. acnes* (−)	*C. acnes* (+)	*p*-Value ^b^ (Mann– Whitney ^b^)
Mobility	Preoperative	2.0	2.0	0.169
	Postoperative (6th month)	1.0	1.0	0.213
		0.008 ^a^	0.025 ^a^	-
Self-Care	Preoperative	1.0	1.0	0.169
	Postoperative (6th month)	1.0	1.0	0.056
		0.083 ^a^	0.157 ^a^	-
Usual Activities	Preoperative	1.5	1.5	0.875
	Postoperative (6th month)	1.0	1.0	1.000
		0.025 ^a^	0.046 ^a^	-
Pain or Discomfort	Preoperative	2.0	3.0	0.591
	Postoperative (6th month)	1.0	2.0	0.825
		0.001 ^a^	0.014 ^a^	-
Anxiety or Depression	Preoperative	2.00	2.00	0.548
	Postoperative (6th month)	1.00	1.00	0.131
		0.008 ^a^	0.025 ^a^	-
EQVas ^c^	Preoperative	80.0	90.0	1.000
	Postoperative (6th month)	30.0	60.0	0.875
	*p*-valor teste Wilcoxon ^a^	0.001 ^a^	0.011 ^a^	-

^a^—Wilcoxon test; ^b^—Mann–Whitney test; ^c^—Visual analogical scale of pain from EQ-5D.

## Data Availability

The research protocol is available at Clinicatrials.gov, and it is also ‘in press’ at an indexed scientific journal [37].

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
