# Peer review of "Would Cutibacterium acnes Be the Villain for the Chronicity of Low Back Pain in Degenerative Disc Disease? Preliminary Results of an Analytical Cohort"

_jpm, 2023, doi:10.3390/jpm13040598_

Round 1
Reviewer 1 Report
this study describes the relationship between low grade infections and LBP. of 265 disc samples, a detection rate of approximately 35% was obtained.
What clinical conclusion would your research have for the patient?
should we not operate on these patients?
Author Response
Dear Reviewer 1,
Below are the answers concerning your questions related to our manuscript:
- Question: Improve conclusions
Answer: The conclusions were improved in the end of the discussions section (lines 343-350, as requested
this study describes the relationship between low grade infections and LBP. of 265 disc samples, a detection rate of approximately 35% was obtained.
What clinical conclusion would your research have for the patient? Should we not operate on these patients?
Answer: Our main conclusion is that the presence of bacteria increases the chances of the patient developing chronic low back pain. Although the mechanisms that explain this finding are still unclear, the use of metabolomics techniques can predict this occurrence through the identification of a molecular profile (metabolomic signature; biomarker) in patients with low back pain. The use of non-invasive methods for identifying metabolomic signatures has already been successfully used in other clinical scenarios and the possibility of identifying them in a patient with discogenic low back pain, for instance, can avoid failures in surgical interventions performed when there is a greater chance of unfavorable evolution due to the chronicity of the pain.
Reviewer 2 Report
Dear authors,
thank you very much for this beautiful and very interesting article. I recommend accepting the article for publication after revision of the article.
1. The "Materials and Methods" section needs to be revised. Please complete information on the planned overall study. You say this is a pre-release of the first results, but what is the overall study design? How many patients are planned, over how long? When do which study-related examinations take place (for example are skin taps of patients performed pre-OP to examine whether there is colonization of the skin with C. acnes)? In addition, a reference to the present ethics vote would be necessary or useful here. Why were patients over 65 years of age excluded?
2. It is essential that the results show how the positive samples are distributed among the patients. Are usually only one or two samples positive? Or rather (almost) all? Since a single pathogen detection of a low-virulent bacterium is often considered contamination in orthopaedic surgery, further processing should definitely be carried out here. This differentiation should necessarily include whether the patients in the preoperative skin smear (which apparently was performed) were colonized with C. acnes.
In Table 4 I do not understand why it is stated: Number of C. acnes positive patients 8. Further ahead in the text is "21 were obtained from colonized patients" (line 161).
3. Throughout the text document, it should always be specified precisely whether "colonized patients" (e.g. line 203) refers to colonization of the intervertebral disc or skin colonization pre-op. This would facilitate understanding.
4. As an orthopaedic surgeon, I would like a more detailed description of the surgical technique, possibly with a small illustration: was there a cage insert in the disc compartment? If so, which one in what way?
Author Response
Dear Reviewer 2,
Below are the answers to your questions concerning our manuscript:
Dear authors,
thank you very much for this beautiful and very interesting article. I recommend accepting the article for publication after revision of the article.
Question: The "Materials and Methods" section needs to be revised. Please complete information on the planned overall study. You say this is a pre-release of the first results, but what is the overall study design? How many patients are planned, over how long?
Answer: We apologize for missing some details described in the material and Methods section, but we have revised and improved by including more descriptive details about the technics used, specially about the isolation and identification of C. acnes (lines 125 to 151). Concerning the pre-released paper with the protocol describing he details of the study, the reference 37 referees to this recent publication (https://doi.org/10.1371/journal.pone.0271773)
Question: When do which study-related examinations take place (for example are skin taps of patients performed pre-OP to examine whether there is colonization of the skin with C. acnes)?
Answer: No, we did not collected a swab pre-OP, but after cleaning the patients’ skin we have collected to make sure that our method was efficient to avoid the contamination with any bacteria. In fact, among all patients included in the study, only 8 was positive for C. acnes found in the skin even after it was cleaned. It demonstrates a low rate of isolation after cleaning.
Question: In addition, a reference to the present ethics vote would be necessary or useful here. Why were patients over 65 years of age excluded?
Answer: They were excluded based on the average life expectancy of our country (70 years), which would imply losses for a follow-up of more than 5 years in the general protocol of the cohort, such as, for example, the evaluation of the chronicity of low back pain in the medium and long term, the evaluation of the transformation of the chronic pain pattern (neuropathic, nociplastic, mixed, etc.), the occurrence of failed-back surgery syndrome and the need for new surgical interventions
Question: It is essential that the results show how the positive samples are distributed among the patients. Are usually only one or two samples positive? Or rather (almost) all? Since a single pathogen detection of a low-virulent bacterium is often considered contamination in orthopaedic surgery, further processing should definitely be carried out here. This differentiation should necessarily include whether the patients in the preoperative skin smear (which apparently was performed) were colonized with C. acnes.
Answer: We apologize about the confusion. Five fragments are collected from each patient’s disc. If two samples are positive in five, it is considered a positive sample. We have used the same criteria from IDSA. As mentioned before, a swab was collected after cleaning the pre-OP site to make sure that this anti-septic procedure would eliminate the presence of C. acnes. Only 8 patients we positive for C. acnes demonstrating the efficacy of this procedure.
Question: In Table 4 I do not understand why it is stated: Number of C. acnes positive patients 8. Further ahead in the text is "21 were obtained from colonized patients" (line 161).
Answer: We completely agree with this comment. We have made another table (Table 4; page5; line: 192), demonstrating the isolation of C. acnes according to the number of patients according to the number of isolates.
Question: 3. Throughout the text document, it should always be specified precisely whether "colonized patients" (e.g. line 203) refers to colonization of the intervertebral disc or skin colonization pre-op. This would facilitate understanding.
Answer: We apologize about that. Along the text we have emphasized if it was the disc content or the patient skin.
Question: 4. As an orthopaedic surgeon, I would like a more detailed description of the surgical technique, possibly with a small illustration: was there a cage insert in the disc compartment? If so, which one in what way?
Answer: We have detailed a little bit the surgical technique. If necessary we can provide a small illustration of the procedure. Briefly, the participants were operated through the posterior midline route, with the goal of neural decompression (laminectomy), discectomy and arthrodesis of levels treated. For segmentar stabilization, pedicular titanium screws connected by rods were used, and interbody PEEK cages were inserted by transforaminal technique (TLIF – transforaminal interbody fusion) in each level treated; a mixture of autologous bone and synthetic graft was used both in the intersomatic spaces and between the transverse processes to improve bone fusion.
Round 2
Reviewer 2 Report
1. In the result part (from line 182) there is a duplication of specified results, this should be removed ("The number of IVD samples analyzed, 21collected from each patient varied from 5-20, with a total of 265 samples and an average of 11.5 samples per patient. In 160 samples (15 patients) were obtained from colonized patients, of which only 18 providednegative. The number of samples collected from each patient ranged from 5 to 20, totaling 265 samples, with an average of 11.5 samples collected per patient. In 160 samples from 15 patients, no positive samples (were found.")
2. Table 4 "Cultibacterium" has to be corrected
Author Response
Dear Reviewer,
We apologize for the mistakes.
The duplication in line 182 was corrected. And concerning the misspelling of the word Cutibacterium in table 4, it was also corrected.